# Evaluating Simulated Climate Patterns from the CMIP Archives Using Satellite and Reanalysis Datasets using the Climate Model Assessment Tool (CMATv1)

John T. Fasullo[1]

[1]National Center for Atmospheric Research, Boulder, CO, 80302, USA

*Correspondence to*: John T. Fasullo (fasullo@ucar.edu)

**Abstract.**

An objective approach is presented for scoring coupled climate simulations through an evaluation against satellite and reanalysis datasets during the satellite era (i.e. since 1979). The approach is motivated, described, and applied to available Coupled Model Intercomparison Project (CMIP) archives and the Community Earth System Model (CESM) Version 1 Large Ensemble archives with the goal of robustly benchmarking model performance and its evolution across CMIP generations. A scoring system is employed that minimizes sensitivity to internal variability, external forcings, and model tuning. Scores are

based on pattern correlations of the simulated mean state, seasonal contrasts, and ENSO teleconnections. A broad range of feedback-relevant fields is considered and summarized on discrete timescales (climatology, seasonal, interannual) and physical realms (energy budget, water cycle, dynamics). Fields are also generally chosen for which observational uncertainty is small compared to model structural differences.

Highest mean variable scores across models are reported for well-observed fields such as sea level pressure, precipitable water, and outgoing longwave radiation while the lowest scores are reported for 500 hPa vertical velocity, net surface energy flux, and precipitation minus evaporation. The fidelity of models is found to vary widely both within and across CMIP generations. Systematic increases in model fidelity in more recent CMIP generations are identified, with the greatest improvements occurring in dynamic and energetic fields. Such examples include shortwave cloud forcing and 500 hPa eddy

geopotential height and relative humidity. Improvements in ENSO scores with time are substantially greater than for climatology or seasonal timescales.

Analysis output data generated by this approach is made freely available online from a broad range of model ensembles, including the CMIP archives and various single-model large ensembles. These multi-model archives allow for an expeditious

analysis of performance across a range of simulations while the CESM large ensemble archive allows for estimation of the influence of internal variability on computed scores. The entire output archive, updated and expanded regularly, can be accessed at: http://webext.cgd.ucar.edu/Multi-Case/CMAT/index.html .

# 1 Introduction

Global climate models were first developed over half a century ago (Hunt et al. 1968, Manabe et al. 1975) and have
provided insight into the climate system on a range of issues including the roles of various physical processes in the climate
system and the attribution of climate events. They also are key tools for near-term initialized prediction and long-term
boundary forced projections. Given their relevance for addressing issues of considerable socioeconomic importance, climate
models are increasingly being looked to for guiding policy-relevant decisions on long timescales and on regional levels.
Many barriers exist however, chief amongst which are the biases in climate model representations of the physical system.

Adequate evaluation of climate models is nontrivial however. A key obstacle is that the longest observational records tend
to monitor temperature and sea level pressure and are therefore not directly related to many of the fields thought to govern
climate variability and change, such as for example cloud radiative forcing and rainfall (Burrows et al. 2018). Global direct
observations of more physically relevant fields exist but are available exclusively from satellite and thus are limited in
duration, with some of the most important data records beginning only in recent decades. Over longer timescales,
uncertainties in forcing external to the climate system (e.g. anthropogenic aerosols) further complicate model evaluation.
Benchmarks of model performance must therefore be designed to deal with associated uncertainties and minimize their
influence.

## 1.1 Motivations

Climate modeling centers continually refine their codes with the goal of improving their models. The Climate Model
Intercomparison Project (CMIP) is an effort to systematically coordinate and release targeted climate model experiments of
high interest in the science community and has thus far provided three major releases, including CMIP3 (Meehl et al. 2007),
CMIP5 (Taylor et al. 2012), and CMIP6 versions (Eyring et al. 2016). Major advances have also recently been made in key
observationally-based climate datasets (as discussed herein). An opportunity has therefore arisen to take stock of these
simulation archives and conduct a retrospective assessment of progress that has been made and challenges that remain.

While individual models are widely scrutinized, systematic surveys of model performance are relatively rare. Evaluation of
single CMIP generations have been conducted and these have been uniquely useful for identifying canonical model biases
(Gleckler 2008, Pincus et al. 2008) It is the goal of this study to provide a similar benchmarking of models, but considerably
expanded in scope in considering multiple CMIP versions and using newly available process-relevant observations that
contextualize model-observation differences with respect to both internal variability and observational uncertainty. An
additional goal is to provide related diagnostic outputs directly to the community. Both the graphical and data outputs
generated may potentially be incorporated into broader community packages such as ESMValTool (Eyring et al. 2020), thus
providing a unique evaluation of fully-coupled physical climate states that encompasses both climatological means and
temporal variations, that accounts for key uncertainties, and that benchmarks models across CMIP generations.

## 1.2 Challenges

A number of challenges exist for efforts aimed at comprehensively assessing climate model fidelity. Observations of many fields that are central to climate variability and change (e.g. cloud microphysics, entrainment rates, aerosol-cloud interactions, Knutti et al. 2010) are not observed on the global, multi-decadal timescales required to comprehensively evaluate models. Fields for which observations do exist often entail uncertainties that are large, particularly at times when the spatial sampling of observing networks is poor (e.g. SST datasets) or for fields that contain significant uncertainty in

satellite-based retrieval (e.g. surface turbulent and radiative fluxes). For instances in which extended data records are unavailable, associated sensitivity to internal variability and externally imposed forcing, which also contains major uncertainties, must be considered, and evaluation of trends are particularly susceptible. In addition, model tuning methods vary widely across centers (e.g. Hourdin et al. 2017, Schmidt et al. 2017), and in instances where climate fields are explicitly tuned, direct comparison against observations is unwarranted.

## 1.3 Approach

The need for objective climate model analysis was highlighted in the 2010 IPCC Expert Meeting on Assessing and Combining Multi-Model Climate Projections (Knutti et al. 2010). Its synthesis report detailed a number of summary recommendations including the consideration of feedback-relevant, process-based fields, and the implementation of metrics that are both simple and statistically robust. In addition, fields were recommended for which observational uncertainty and

internal variability are both quantifiable and small relative to model structural differences. The reliance on any single evaluation dataset was also deemed problematic in that doing so might be both susceptible to compensating errors and insufficient to fully characterize inter-model contrasts. The approach here is guided, in part, by these recommendations.

Various model analysis efforts have focused on surface temperature (e.g. Braverman et al. 2017, Lorenz et al. 2019). A

thorough evaluation of climate model thermodynamics is provided by the TheDiaTo as described in Lembo et al. (2019). Complex measures of model performance that allow for a richer comparison of model statistics against observations have also been discussed (Gibbs and Su 2002) and proposed (Ghil 2015). The approach adopted here highlights instead the main components of the energy and water cycles using simple diagnostic measures. Objective assessments of CMIP3 performance based on the mean climate state using these fields were performed in Gleckler et al. (2008) and Pincus et al. (2008). The goal

of this work is to complement and extend these efforts in including an analysis of both the mean state and variability across three generations of CMIP simulations while distinguishing between timescales and realms of diagnostics, and using improved observational datasets and constraints (described below). As the skill of a given climate model is likely to depend on the relevant application (Gleckler et al. 2008; Pierce et al. 2009, Knutti et al. 2017), the scores computed herein are made widely available to the community and may help guide formation of optimal model subsets for targeted applications.


The consideration of multiple CMIP generations is motivated in part by reported shifts in model behavior, such as for example the apparent increase in climate sensitivity to carbon dioxide in some models (Gettelman et al. 2019, Golaz et al. 2019, Neubauer et al. 2019). Do such shifts accompany systematic improvements in models and if so, in what fields? It is also of a more general interest to quantify canonical biases in models, their changes in successive model generations, and

persistent biases affecting the most recent generations of climate models. The specific questions addressed here therefore include: what improvements have occurred across model generations and what persistent biases remain? What process-relevant well-observed fields are models most skillful in reproducing? To what extent are apparent improvements and persisting biases robustly detectable in the presence of internal climate variability, particularly as they relate to brief satellite records?


**2.0 Methods**

The analysis approach consists of computing a range of scores based on pattern correlations encompassing three climatic timescales: the climatological annual mean (annual), seasonal mean contrasts (JJA-DJF), and ENSO teleconnection patterns-

computed from the 12-month July through June mean regressions against Niño3.4 sea surface temperatures (SST). The choice of ENSO as a model diagnostic is motivated in part by the demands involved in its accurate simulation arising from the highly coupled nature of the mode; which includes feedbacks between clouds, diabatic heating, and winds in the atmosphere, and currents and steric structure in the ocean (e.g. Cheng et al. 2018). Variables are classified according to three variable types (or realms) corresponding to the energy budget, water cycle, and dynamics. To reduce the influence of internal

variability, the time period over which these fields are considered is at least 20 years, though the availability of some datasets allows for the use of longer periods, further reducing the susceptibility of the analysis to internal variability. Contemporaneous time intervals are also chosen to provide for maximum overlap between observed and simulated fields. The variables selected for consideration are chosen based on availability and judgment of their importance in simulating climate variability and change. In part this judgment is based on a recent community solicitation (Burrows et al. 2018) and

some of the fields included (e.g. TOA fluxes) are deemed by experts to be optimal metrics for model evaluation (e.g. Baker and Taylor 2016).

*2.1 Observational Datasets*

*The Energy Budget Realm*
Energy budget fields considered consist broadly of TOA radiative fluxes and cloud forcing, vertically integrated atmospheric energy divergence and tendency, and surface heat fluxes. Radiative fluxes at TOA are taken from the Clouds and Earth's Radiant Energy System (CERES) Energy Balance and Filled Version 4.1 dataset (EBAFv4.1, Loeb et al. 2018). The dataset

offers a number of improvements over earlier versions and datasets, with improved angular distribution models and scene
identification, but is perhaps most notable for its recently updated derivation of cloud radiative forcing (CF). Historically CF
has been estimated from observations by differencing cloudy and neighboring clear regions, with the effect of aliasing
meteorological contrasts between the regions (whereas models merely remove clouds from their radiative transfer scheme
using collocated meteorology). In the EBAFv4.1, fields from NASA's GEOS-5 reanalysis (Borovikov et al. 2019) are used
to estimate fluxes and CF for collocated (rather than remote) atmospheric conditions, thus providing for a more analogous
comparison to models.  From CERES, the TOA net shortwave (ASR), outgoing longwave (OLR), and net ($R_T$) radiative
fluxes are used. In addition, estimates of shortwave CF ($SW_{CF}$) and longwave CF ($LW_{CF}$) are used.

Derived from the ERA-Interim reanalysis (Dee et al. 2011), vertical integrals of atmospheric energy are used to both assess
the total energy divergence within the atmosphere ($\nabla \cdot A_E$) and its tendency ($\partial A_E / \partial t$). This provides important insight into
the regional generation of atmospheric transports and their cumulative influence on the global energy budget (e.g. Fasullo
and Trenberth 2008). They are also an energy budget component necessary for computing the net surface energy fluxes from
the residual of $R_T$, $\nabla \cdot A_E$ , and $\partial A_E / \partial t$. Given the challenges of directly observing the net surface flux, a residual method is
likely the best available method for estimating the large-scale evaluation of the surface heat budget. The method has been
demonstrated to achieve an accuracy on par with direct observations on regional scales and has proven superior on large
scales, where the atmospheric divergences on which they rely become small, converging to zero by definition in the global
mean (Trenberth and Fasullo, 2017). Uncertainty estimation of CERES fluxes is also well documented (Loeb et al. 2018).

*The Water Cycle Realm*

Water cycle fields considered include precipitation (P), evaporation minus precipitation (EP), precipitable water (PRW),
evaporation (LH), and near-surface relative humidity ($RH_S$). The utility of P and EP as model diagnostics was highlighted by
Greve et al. (2018) in selecting a subset of CMIP5 models. As global evaporation fields from direct observations and
estimated from satellite also contain substantial uncertainty, precipitation minus evaporation is estimated here instead from
the vertically integrated divergence of moisture simulated in ERA-Interim fields, which is also arguably the most accurate
means of evaluating large scale patterns and variability (Trenberth and Fasullo, 2013). Precipitation is estimated from the
Global Precipitation Climatology Project (Huffman et al. 2013) Climate Data Record (Adler et al. 2016). The improved
version takes advantage of improvements in the gauge records used for calibration and indirect precipitation estimation from
longwave radiances provided by NOAA leo-IR data. For other water cycle fields, output from the European Centre for
Medium Range Weather Forecasts (ECMWF) Reanalysis Version 5 (ERA5, Hersbach et al. 2019) is used. ERA5 is the
successor to ERA-Interim, increasing the resolution of reported fields, the range of fields assimilated from satellite
instruments, and the simulation accuracy as compared against a broad range of observations for various measures. For

example, a comparison of ERA5 to satellite data (CERES, GPCP) demonstrates reduced mean state annual and seasonal biases as compared to ERA Interim (not shown).

*The Dynamical Realm*

Dynamical fields considered include sea level pressure (SLP), wind speed ($U_s$), 500 hPa eddy geopotential height ($Z_{500}$), vertical velocity ($W_{500}$), and relative humidity ($RH_{500}$). The use of eddy geopotential rather than total geopotential, which contains significant spatial variance arising from meridional temperature contrasts, is motivated by its ability to resolve our main field of interest - the spatial structure of atmospheric circulations. ERA5, discussed above, is used for estimation of dynamical fields, as such fields are generally not provided from satellite (excepting $RH_{500}$). Motivating its use, and among its

notable improvements relative to earlier reanalyses, is ERA5's improved representation of tropospheric waves and jets that is core to our dynamical evaluation.

*2.2 Generation of Variable, Realm, Timescale, and Overall Scores*

Scores for annual mean, seasonal mean, and ENSO timescale metrics are generated from the area-weighted pattern

correlations (Rs) between each simulated variable and the corresponding observational dataset. Weighted averages of these three Rs are then used to generate a Variable Score for each field in a given simulation. Arithmetic averages across the relevant Variable Scores are then used to generate Realm Scores, and the Realm Scores for a simulation are arithmetically averaged to generate an Overall Score. Similarly, Timescale Scores are generated by averaging Rs for the relevant timescale across all variables. The inclusion of both Realm and Timescale scores is motivated in part by the need to interpret the origin

of changes in Overall Scores, which include a large number of Rs that may otherwise obscure an obvious physical interpretation. Insights gained, for example, include the attribution of much of the Overall Score improvement across CMIP generations to the fidelity of simulated ENSO patterns.

The use of weights in generating Variable Scores is motivated by the desire to assist in interpretation of differences in the

Overall Score relative to the influence of internal variability. Using the Community Earth System Version 1 Large Ensemble (CESM1-LE, Kay et al. 2015), weights for ENSO scores are reduced from 1.000 to 0.978 (while for annual and seasonal scores they are 1.000) such that the standard deviation range in Overall Scores for the 40 members of the CESM1-LE is 0.010. This therefore can be used to interpret generally the approximate contribution of internal variability to inter-model Overall Scores in analysis of the CMIP archives, suggesting that differences between individual simulations of less than

approximately 0.040 ($\pm 2\sigma$) are not statistically significant. Where available, multiple-simulation analyses provide an opportunity for further narrowing the uncertainty of statements regarding inter-model fidelity, and as will be seen, Overall Score ranges within and across the CMIP ensembles generally exceed the obscuring effects of internal variability.

*2.3 CMIP Simulations*

As the goal of this work is to characterize the evolution of agreement between climate models generally across the CMIP archives, and observations, all available model submissions for which sufficient data are provided are included in the analysis (as summarized in Table 1). A major exception to the data availability requirement relates to near surface wind speed ($U_S$), which was not included as part of the CMIP3 variable list specification. Scores for the dynamical realm in CMIP3 therefore omit $U_S$ as a scored variable and instead compute the dynamic Realm score from the remaining dynamic

variable scores. While multiple ensemble members are provided in the CMIP archives for many models, and have been assessed, only a single member of each model is incorporated into the analysis here to avoid overweighting the influence of any single mode.

    Lastly, in an effort to quantify the leading patterns of bias that differentiate models, a covariance matrix based principal

component (PC) analysis is used where the array of bias patterns (lon x lat x model) is decomposed for its empirical orthogonal functions (EOFs). The EOFs are plotted as regressions against the normalized PC timeseries and therefore have the same units as the raw fields. Shown are the two leading EOFs and corresponding PC values, sorted by their values and averaged across terciles for each CMIP generation. Included in the PC analysis is an observational estimate (i.e. zero bias) to provide context for model differences. The leading EOFs are found to be both separable and explain significant variance in

the bias matrix.

## 3.0 Assessing CMIP Scores

    To illustrate the analysis approach and provide context for the magnitude of biases relative to internal variability and

observational uncertainty, Figure 1 shows both observed and simulated $SW_{CF}$ fields across the timescales considered (Fig. 1a, annual, 1b) seasonal, and 1c) ENSO) in the CESM Version 2 submission to CMIP6, CERES estimates (Fig. 1d-f), and their differences (CESM2-CERES, Fig. 1g-i). Significant spatial structure characterizes all fields, with a strong $SW_{CF}$ cooling influence in the mean across much of the globe (Fig. 1a), seasonal contrasts (Fig. 1b) that vary between land and ocean and latitudinal zone, and ENSO teleconnections (Fig. 1c) that extend from the tropical Pacific Ocean to remote ocean

basins and the extratropics. While (as will be seen), CESM2 scores among the best available climate models, large model-observation differences nonetheless exist. Regions where model-observation differences are larger than twice the ensemble standard deviation in the CESM1-LE in the annual and seasonal means (stippled) are widespread and remain extensive where the uncertainty range is expanded to incorporate estimated observational uncertainty (added in quadrature, hatched) from Loeb et al. 2018. Of particular note is the fact that it is the large-scale coherent patterns of bias, where model-

observational disagreement exceeds uncertainty bounds, that are the primary drivers of pattern correlations used in scoring, rather than synoptic scale noise.

The color table summary of scores for CMIP3 (mean pattern correlations scaled by 100, Figure 2) provides a visual summary of simulation performance across the models in the archive (abscissa), including Variable, Realm, Timescale, and Overall Scores (i.e. aggregate scores, ordinate). Simulations are sorted by Overall Scores (top row, descending scores toward right). Realm and Timescale Scores (rows 2 through 7) also provide broad summaries of model performance. Mean Overall Scores (69±7, 1 sigma) are modest generally in CMIP3 and generally uniform across realms. CMIP3 simulations score particularly poorly for ENSO, where scores average to 47, are generally less than 60, and approach 0 in some coarse-grid models. Variable scores are highest for PRW and OLR (which are strongly tied to surface temperature), and for SLP, and less for other variables, with the lowest scores reported for $R_S$ and W500. Spread across models for $R_S$ is particularly large relative to other variables. Average variable scores are also poor for $SW_{CF}$ (68), $LW_{CF}$ (71), and P (69), which are among the more important simulated fields according to expert consensus (Burrows et al. 2018).

The color table summary of scores for CMIP5 (Figure 3) reveals scores that are considerably higher than most CMIP3 simulations, with improvements in the average Overall Score of (75±5) and most notable improvements on the ENSO timescale, with an average of 57, though with considerable inter-model range ($\sigma=10$). A broad increase in scores in the highest performing models is apparent with numerous variable scores exceeding 85 (orange/red) and several Overall Scores of 80 or better. As for CMIP3 the highest scoring variables are PRW, SLP, and OLR, while $RH_S$ and $W_{500}$ are among the lowest scoring variables. Mean variable scores remain relatively low for $SW_{CF}$ (71), $LW_{CF}$ (75), and P (73).

The color table summary of scores for CMIP6 (Figure 4) illustrates scores that are considerably higher than both CMIP3 and CMIP5 simulations, with improvements in the average Overall Score of (79±4) and most continued improvements on the ENSO timescale, though again with considerable inter-model range. A continued increase in scores in the highest performing models is again apparent, with scores reaching the mid- to upper 70s and numerous variable scores exceeding 90 (red). The highest scoring variables again include PRW, SLP, and OLR though scores are also high for $RH_{500}$, one of the more important simulated fields according to expert consensus (Burrows et al., 2018). Scores also increase for $SW_{CF}$ (78), $LW_{CF}$ (80), and P (77).

To highlight connections between variables, and aid in identifying the main variables driving variance in aggregate scores across the CMIP archives, correlations amongst scores across all CMIP models are shown in Figure 5. For Overall Scores, these include strong connections to P, E-P and OLR, fields strongly connected to atmospheric heating, dynamics, and deep convection and therefore broadly relevant to model performance. Strong connections also exist for $SW_{CF}$, $LW_{CF}$, and $RH_{500}$, consistent with the expert consensus in highlighting these fields as being particularly important (Burrows et al. 2018). An approximately equal correlation exists across Realms with the Overall Score, while for timescales, ENSO exhibits the strongest overall correlation as it contains the greatest inter-model variance and thus explains a greater portion of the Overall

Score variance. Correlations between timescales is weak generally, consistent with the findings of Gleckler et al. (2008) where relationships were also examined between the mean state and interannual variability. Notable as well is that some variables for which scores are high in the mean, such as SLP and PRW, exhibit little correlation with the Overall Score as the uniformly high scores across models impart relatively little variance to the Overall Scores.


## 4.0 Derived Bias Patterns for Selected Variables

The observational estimate of $SW_{CF}$ from CERES is shown in Figure 6a along with mean bias patterns for CMIP3 (b) and CMIP6 (c). A principal component (PC) analysis of the bias across the broader CMIP archives is also conducted (see

Methods) with the leading principal components and their tercile mean values within each CMIP version being shown (d) along with the two leading patterns of bias (Fig. 6e, f). The mean observational field (Fig. 6a) is characterized by negative values in nearly all locations (except over ice) and the strongest cooling influence in the deep tropics, subtropical stratocumulus regions, and midlatitude oceans. Mean bias patterns demonstrate considerable improvement across the CMIP generations, with major reductions in negative biases in the subtropical and tropical oceans. Variance across models is

characterized by the degree of tropical-extratropical contrasts in $SW_{CF}$ (EOF1), which explains 24% of the inter-model variance, and land-ocean contrasts (EOF2), which explain 16% of the variance. The expression of both patterns of biases is demonstrated to diminish across CMIP generations and terciles in their PC weights (Fig. 6d), ordered sequentially (1-3) with CMIP6 values (dark blue) lying generally closer to observations than CMIP3/5. Improvements are not in however necessarily monotonic across the CMIP generations, with improvements and degradations notable in some aspects of the

PC1/2 transition from CMIP3 to CMIP5 (i.e. instances in which tercile mean PC values are closer to CERES for CMIP3 than CMIP5).

The observational estimate for $LW_{CF}$ from CERES is shown in Figure 7a along with mean bias patterns for CMIP3 (b) and CMIP6 (c). A PC analysis of the bias across the CMIP archives is also shown with the leading PC weights and their tercile

mean values within each CMIP version being shown (d) along with the two leading patterns of bias (Fig. 7e, f). Observational fields are characterized by a strong heating influence in regions of deep tropical convection and in the extratropical ocean regions in which $SW_{CF}$ is also strong (Fig. 6a) while weak heating is evident in the subtropics and polar regions. Significant changes characterize mean bias patterns between CMIP3 and CMIP6, with positive biases across most ocean regions in CMIP3 and negative biases in many of the same regions in CMIP6. On average however, the magnitude of

biases are reduced across CMIP generations. This is evident for example in the PC analysis of bias (Fig. 7d), where CMIP6 values lie closer generally to CERES than for CMIP3 or CMIP5. The leading mode (EOF1, Fig. 7e) exhibits strong weightings over the warm pool, is negatively correlated with both the mean pattern and bias, and explains 36% of the inter-model variance. In contrast, EOF2 exhibits a strong tropical-extratropical contrast, little correlation to the mean pattern or

bias, and explains only 13% of the variance. The PC1/2 tercile weights for these modes show a considerable reduction in
EOF1 spread, smaller mean tercile biases generally, and improved agreement across model terciles from CMIP3 to CMIP6,
though as with SW$_{CF}$, the improvement is not monotonic nor uniform across all terciles and PCs.

The observational estimate for precipitation from GPCP is shown in Figure 8a along with mean bias patterns for CMIP3
(Fig. 8b) and CMIP6 (Fig. 8c). The PC analysis of the bias across the CMIP archives is also shown with the leading PC
tercile mean values for each CMIP version being shown (Fig. 8d) along with the two leading patterns of bias (Fig. 8e, f). The
annual mean pattern resolves key climate system features, including strong precipitation in the Inter-Tropical Convergence
Zone (ITCZ) and arid conditions in the subtropics and at high latitudes. Biases are large in both CMIP3 and CMIP6 on
average and are characterized generally by excessive subtropical precipitation and deficient precipitation in the Pacific
Ocean ITCZ, South America, and at high latitudes. Earlier work has generally characterized model bias in terms of its double
ITCZ structure (Oueslati et al. 2015), though systematic bias is also apparent beyond the tropical Pacific Ocean. In addition,
the PC decomposition of CMIP precipitation biases (Fig. 8d-f) suggests that the bias is comprised to two orthogonal leading
patterns that together explain 15% and 11% of the variance across models, respectively. A separable unique leading pattern
is therefore not evident. Rather, the leading pattern (Fig. 8e) is characterized by weakness in precipitation across the
equatorial oceans, with elevated rates in the Maritime continent and in the Pacific Ocean near 15N/S. The second pattern
(Fig. 8f) is characterized by loadings over Africa and South America, and on the southern fringe of the observed
climatological Pacific ITCZ (Fig. 8a), with negative loadings in the subtropical ocean basins. Based on mean PC tercile
values, slight improvement across CMIP generations is evident, as tercile values lie closer to observations for all terciles of
PC1/2 in CMIP6 versus CMIP3, with the exception of the first tercile of PC1, where CMIP3 lies close to GPCP.

The observational estimate for RH$_{500}$ from ERA5 is shown in Figure 9a along with mean bias patterns for CMIP3 (b) and
CMIP6 (c). A principal component analysis of the bias across the CMIP archives is also shown with the leading principal
components and their tercile mean values within each CMIP version being shown (Fig. 9d) along with the two leading
patterns of bias (Figs. 9e, f). The observed RH$_{500}$ field is characterized by positive humidity biases in regions of frequent
deep convection (i.e. Maritime Continent, Amazon) and at high latitudes, and very dry conditions in the subtropics, with
values generally below 30% across the subtropics, features that were poorly resolved in CMIP3 (e.g. Fasullo and Trenberth
2012). The CMIP3 mean bias field is negatively correlated with the mean state, with patterns that lack sufficient spatial
contrast, are too moist in the subtropics, and too dry in Africa, the Maritime continent, the Amazon, and at high latitudes.
The magnitude of mean RH$_{500}$ biases in CMIP6 are substantially smaller (roughly 50%) than CMIP3, though they share a
similar overall pattern reflecting weakness in spatial contrasts. The PC analysis of bias reveals a leading pattern that explains
50% of the intermodal variance and is negatively correlated with observations (-0.44). The second leading pattern (Fig. 9f)
explains considerably less variance (15%) and exhibits a zonally uniform structure characterized by tropical-extratropical

contrast. The weights for PC1/2 reveal systematic bias in PC1 across models (all lie to the right of ERA5), and considerable improvement across CMIP generations as CMIP6 weights lie significantly closer to ERA5 that CMIP3 weights for all terciles (1-3). Small improvements are also evident in terciles 1 and 2 of PC2, though this comprises a small fraction of variance in overall CMIP bias.

In the effort to summarize the evolution of the full distributions of scores across the CMIP archives, whisker plots encompassing the median, interquartile, and $10^{th}$-$90^{th}$ percentile ranges are shown for various aggregate metrics and key fields in Figure 10. Also shown are the equivalent ranges for scores computed from the CESM1-LE to provide an estimate of the influence of internal variability for each distribution. A steady improvement in the Overall Scores is evident across CMIP versions, a progression that is also evident across Realm Scores and particularly for the poorest scoring models in the Dynamics Realm. Scores for Annual and Seasonal timescales are generally high across archives, though internal variability is also small and is substantially less than the median improvements across the archives. The range of scores for ENSO is significantly greater than other timescales, as is the range of internal variability, and substantial improvements have been realized for the lowest scoring models across successive CMIP generations. Noteworthy are the substantial improvements in $SW_{CF}$, $LW_{CF}$, and P, with the best CMIP3 simulations scoring near the median value for CMIP6 and improvements in median values from CMIP3 to CMIP6 exceeding uncertainty arising from internal variability. Scores for $RH_{500}$ have also improved, although the spread within the CMIP3 archives is substantial and uncertainty arising from internal variability is somewhat greater than for other variables. $RH_{500}$ scores in CMIP6 are generally higher than for cloud forcing and P. For SLP, median scores are uniformly high across the CMIP generations, with small but steady improvement in median and interquartile scores, with the main exception of high scores being the low scoring 0-25% range of CMIP3 simulations.

## 5.0 Discussion

An objective model evaluation approach has been developed that uses feedback-relevant fields and takes advantage of recent expert elicitations of the climate modeling community and advances in satellite and reanalysis datasets. In its application to the CMIP archives, the analysis is shown to provide an objective means for computing model scores across variables, realms, and timescales. Visual summaries of model performance across the CMIP archives are also generated, which readily allow for the survey of a broad suite of climate performance scores. As there is unlikely to be a single model best-suited to all applications (Gleckler et al. 2008, Knutti et al. 2010, 2017), in providing online access to model scores and the fields used to compute them, the results herein are intended to aid the community in informing model ensemble optimization for targeted applications.

Based on the pattern correlation approach adopted, a number of statements can be made regarding the overall performance of climate models across CMIP generations. Noteworthy is that, as informed by analysis of the CESM1-LE and consistent with the design of the approach used, these statements are robust to the obscuring influence of internal climate variability. In general, computed scores have increased steadily across CMIP generations, with improvements exceeding the range of internal variability. Associated with these improvements, the leading patterns of bias across models are shown to have been reduced. Improvements are large and particularly noteworthy for ENSO teleconnection patterns, as the poorest scoring models in each CMIP generation have improved substantially. In part this may be due to the elimination of very low resolution models in CMIP5/6, though improvements in model physics is also likely to play a role. The overall range of model performance within CMIP versions has also decreased in conjunction with increases in median scores, as improvement in the worst models has generally outpaced that of the median. Reductions in systematic patterns of bias (e.g. Figs. 6-9) across the CMIP archives have been pronounced for fields deemed in expert solicitations to have particular importance, including $SW_{CF}$, $LW_{CF}$, and $RH_{500}$.

Also relevant for climate feedbacks, Variable Scores for $SW_{CF}$, $LW_{CF}$, $RH_{500}$, and precipitation have increased steadily across the CMIP generations (e.g. Fig. 10), with magnitudes exceeding the uncertainty associated with internal variability. Scores are particularly high for CMIP6 models for which high climate sensitivities have been reported, including CESM2, SAM0-UNICON, GFDL-CM4, CNRM-CM6-1, E3SM, and EC-Earth3-Veg (though exceptions also exist such as in the case of MIRCO6). These findings therefore echo the concerns voiced in Gettelman et al. 2019: "What scares us is not that the CESM2 ECS is wrong (all models are wrong, (Box, 1976)) but that it might be right.". The fields provided by CMAT allow for an expedited analysis of the sources of these improvements, such as for example the simulation of supercooled liquid clouds (e.g. Kay et al. 2016). Further work examining the ties between metrics of performance in simulating the present-day climate, such as those provided here, and longer-term climate model behavior is warranted to bolster confidence in model projections of climate change.

**Code and Data Availability**

Data used in this study are available freely from the Earth System Grid at: https://www.earthsystemgrid.org

NetCDF output for the fields generated herein is freely available at: http://webext.cgd.ucar.edu/Multi-Case/CMAT/index.html. The software used to develop this work is available under a free software license at Zenodo: https://zenodo.org/record/3922308 .

**Acknowledgements**

This material is based upon work supported by the National Center for Atmospheric Research, which is a major facility sponsored by the National Science Foundation under Cooperative Agreement No. 1852977. The efforts of Dr. Fasullo in this work were supported by NASA Award 80NSSC17K0565, by NSF Award #AGS-1419571, and by the Regional and Global Model Analysis (RGMA) component of the Earth and Environmental System Modeling Program of the U.S. Department of Energy's Office of Biological & Environmental Research (BER) via National Science Foundation IA 1844590.

**Author Contribution**

JF designed the analysis routine, performed the model analysis, developed the anlaysis, obtained the data from the ESG and created all graphics. JF composed the manuscript.

**Competing Interests**

The author declares that he has no conflict of interest.

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

**Tables**

Table 1: Sorted summary of CMIP models considered in this work, sorted by Overall Scores.

500

| CMIP3 | CMIP5 | CMIP6 |
|---|---|---|
| gfdl_cm2_0 (0.78) | CESM1-BGC (0.81) | CESM2 (0.86) |
| gfdl_cm2_1 (0.75) | CNRM-CM5-2 (0.81) | MIROC6 (0.85) |
| cccma_cgcm3_1_t63 (0.75) | CESM1-FASTCHEM (0.81) | CESM2-WACCM (0.85) |
| mri_cgcm2_3_2a (0.75) | CESM1-CAM5 (0.81) | GISS-E2-1-H (0.85) |
| mpi_echam5 (0.75) | ACCESS1-0 (0.81) | SAM0-UNICON (0.84) |
| miub_echo_g (0.74) | NorESM1-ME (0.80) | GFDL-CM4 (0.84) |
| csiro_mk3_5 (0.74) | CESM1-WACCM (0.80) | EC-Earth3-Veg (0.84) |
| ingv_echam4 (0.73) | CESM1-CAM5-1-FV2 (0.80) | EC-Earth3 (0.83) |
| ukmo_hadcm3 (0.73) | MIROC5 (0.80) | UKESM1-0-LL (0.82) |
| cccma_cgcm3_1 (0.73) | CMCC-CMS (0.80) | MRI-ESM2-0 (0.82) |
| cnrm_cm3 (0.73) | HadGEM2-ES (0.80) | E3SM-1-0 (0.81) |
| ncar_ccsm3_0 (0.72) | NorESM1-M (0.79) | CNRM-CM6-1 (0.81) |
| csiro_mk3_0 (0.71) | BNU-ESM (0.79) | CNRM-ESM2-1 (0.81) |
| miroc3_2_medres (0.71) | ACCESS1-3 (0.78) | MIROC-ES2L (0.81) |
| bccr_bcm2_0 (0.71) | HadGEM2-AO (0.78) | FGOALS-g3 (0.79) |
| iap_fgoals1_0_g (0.69) | bcc-csm1-1-m (0.77) | CAMS-CSM1-0 (0.79) |
| miroc3_2_hires (0.69) | GFDL-CM2p1 (0.76) | BCC-CSM2-MR (0.77) |
| ukmo_hadgem1 (0.68) | CanESM2 (0.76) | BCC-ESM1 (0.77) |
| ipsl_cm4 (0.67) | CMCC-CESM (0.75) | CanESM5 (0.77) |
| ncar_pcm1 (0.61) | IPSL-CM5B-LR (0.75) | IPSL-CM6A-LR (0.74) |
| inmcm3_0 (0.60) | MRI-ESM1 (0.75) | GISS-E2-1-G (0.74) |
| giss_model_e_r (0.60) | MPI-ESM-LR (0.75) | NorESM2-LM (0.74) |
| giss_aom (0.59) | MPI-ESM-MR (0.74) | |
| giss_model_e_h (0.46) | MPI-ESM-P (0.74) | |
| | MRI-CGCM3 (0.74) | |
| | FGOALS-g2 (0.74) | |
| | GFDL-ESM2G (0.72) | |
| | GISS-E2-R-CC (0.72) | |
| | IPSL-CM5A-MR (0.71) | |
| | MIROC-ESM (0.70) | |
| | GISS-E2-H-CC (0.69) | |
| | IPSL-CM5A-LR (0.68) | |
| | CSIRO-Mk3-6-0 (0.68) | |
| | MIROC-ESM-CHEM (0.68) | |
| | inmcm4 (0.68) | |
| | GISS-E2-H (0.67) | |
| | CESM1-BGC (0.81) | |
| | CNRM-CM5-2 (0.81) | |
| | CESM1-FASTCHEM (0.81) | |
| | CESM1-CAM5 (0.81) | |
| | ACCESS1-0 (0.81) | |
| | NorESM1-ME (0.80) | |
| | CESM1-WACCM (0.80) | |
| | CESM1-CAM5-1-FV2 (0.80) | |
| | MIROC5 (0.80) | |
| | CMCC-CMS (0.80) | |
| | HadGEM2-ES (0.80) | |
| | NorESM1-M (0.79) | |
| | BNU-ESM (0.79) | |
| | ACCESS1-3 (0.78) | |
| | HadGEM2-AO (0.78) | |
| | bcc-csm1-1-m (0.77) | |
| | GFDL-CM2p1 (0.76) | |
| | CanESM2 (0.76) | |
| | CMCC-CESM (0.75) | |
| | IPSL-CM5B-LR (0.75) | |
| | MRI-ESM1 (0.75) | |
| | MPI-ESM-LR (0.75) | |
| | MPI-ESM-MR (0.74) | |
| | MPI-ESM-P (0.74) | |
| | MRI-CGCM3 (0.74) | |
| | FGOALS-g2 (0.74) | |
| | GFDL-ESM2G (0.72) | |
| | GISS-E2-R-CC (0.72) | |
| | IPSL-CM5A-MR (0.71) | |
| | MIROC-ESM (0.70) | |
| | GISS-E2-H-CC (0.69) | |
| | IPSL-CM5A-LR (0.68) | |
| | CSIRO-Mk3-6-0 (0.68) | |
| | MIROC-ESM-CHEM (0.68) | |
| | inmcm4 (0.68) | |
| | GISS-E2-H (0.67) | |

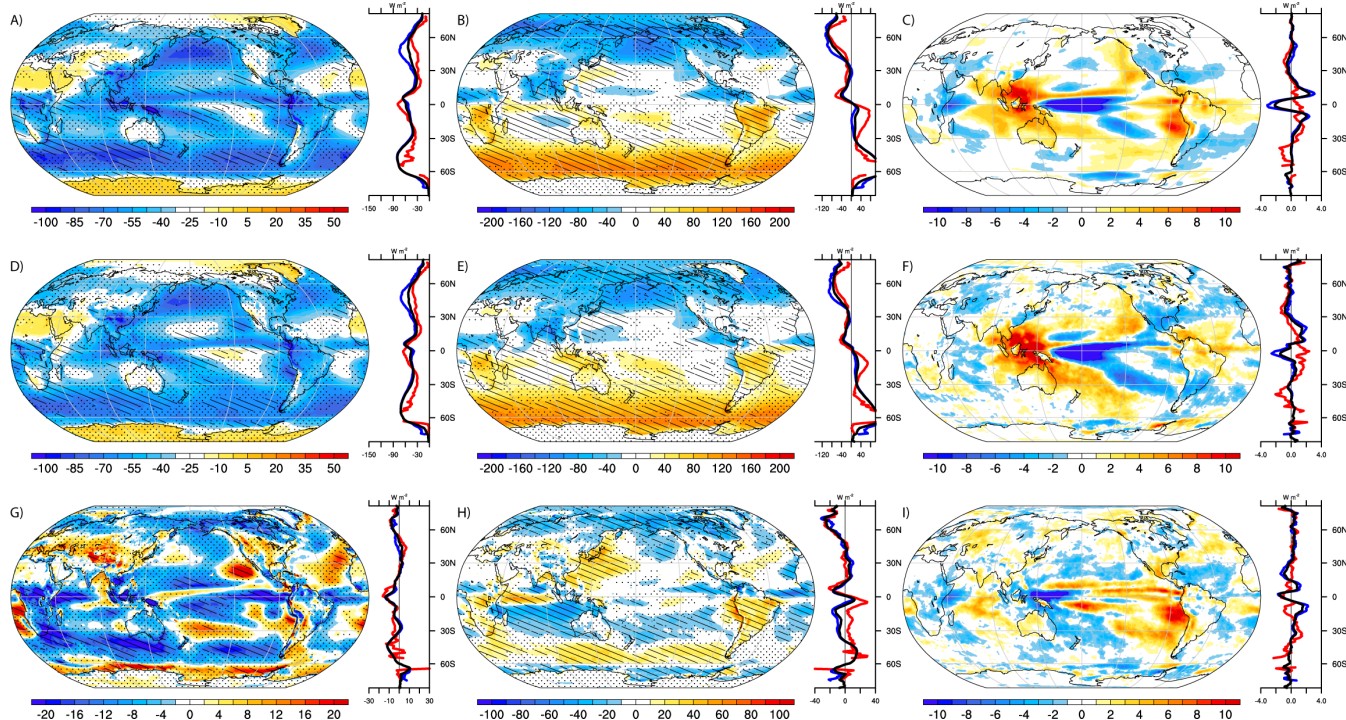

505

**Figure 1: Mean simulated fields of SW$_{CF}$ in CESM2 from 1995-2014 for A) the annual mean, B) seasonal contrasts, and C) regressed against Niño3.4 SST anomalies using July through June averages. Observed CERES EBAF4.1 estimated SW$_{CF}$ for 2000-2018 for analogous metrics (D-F) and CESM2-CERES differences (G-I) are also shown. Stippling indicates regions where CESM2-CERES differences exceed twice the estimated internal spread from CESM1-LE. Hatching indicates regions where differences**

510 **exceed the same spread plus observational uncertainty (added in quadrature, applied to all panels in each column). Units are W m$^{-2}$ except for regressions (right column) where units are W m$^{-2}$ K$^{-1}$. Zonal means (right panels) include land (red), ocean (blue), and global (black).**

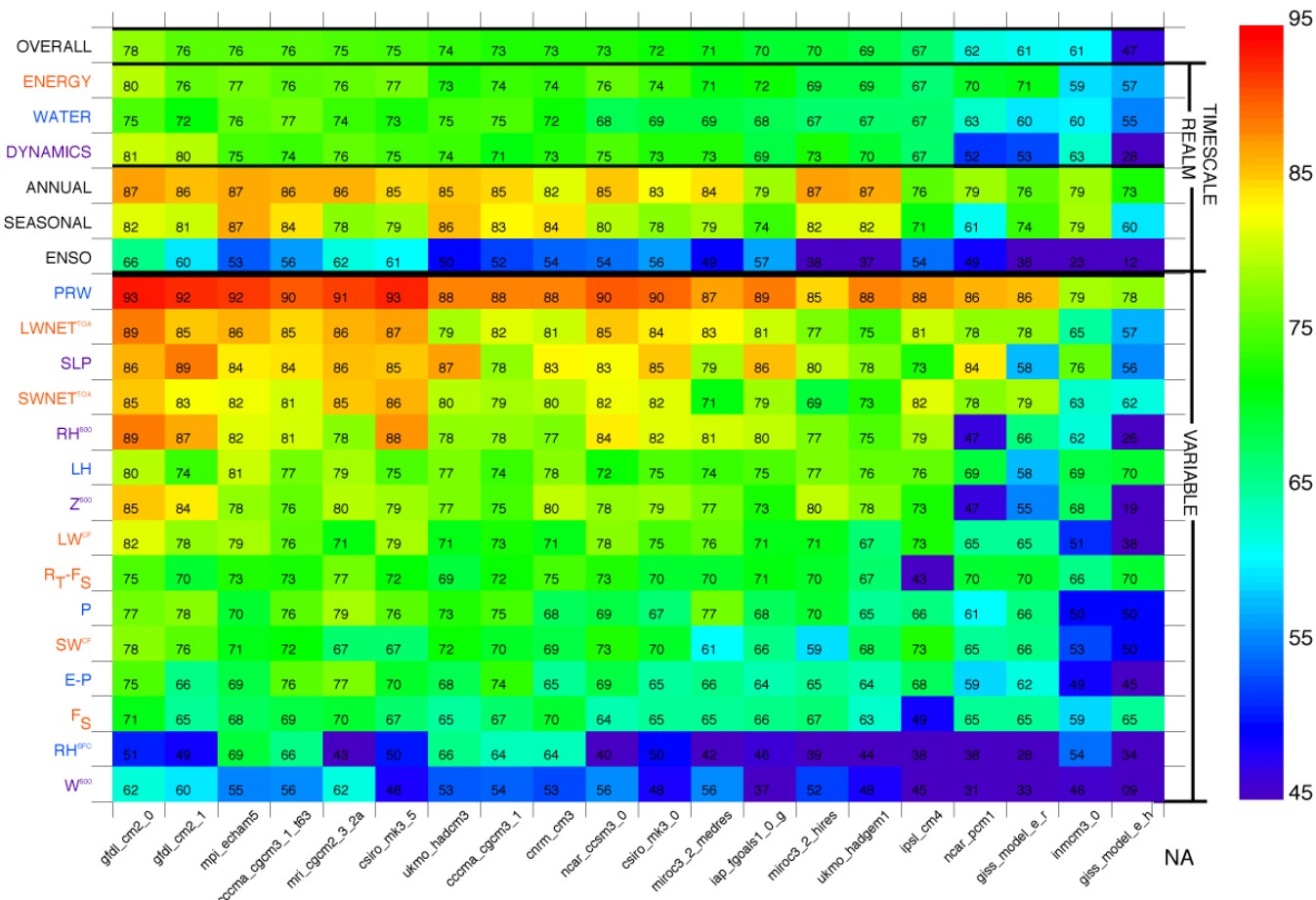

Figure 2: Overall, Realm, Timescale, and Variable scores (ordinate) for historical (20c3m) simulations submitted to the CMIP3 archives (abscissa) sorted by overall score (top row) based on methods employed (see Methods). Simulations and variables are ordered in descending score order from left to right using the Overall Score and from top to bottom using average Variable Score, respectively.

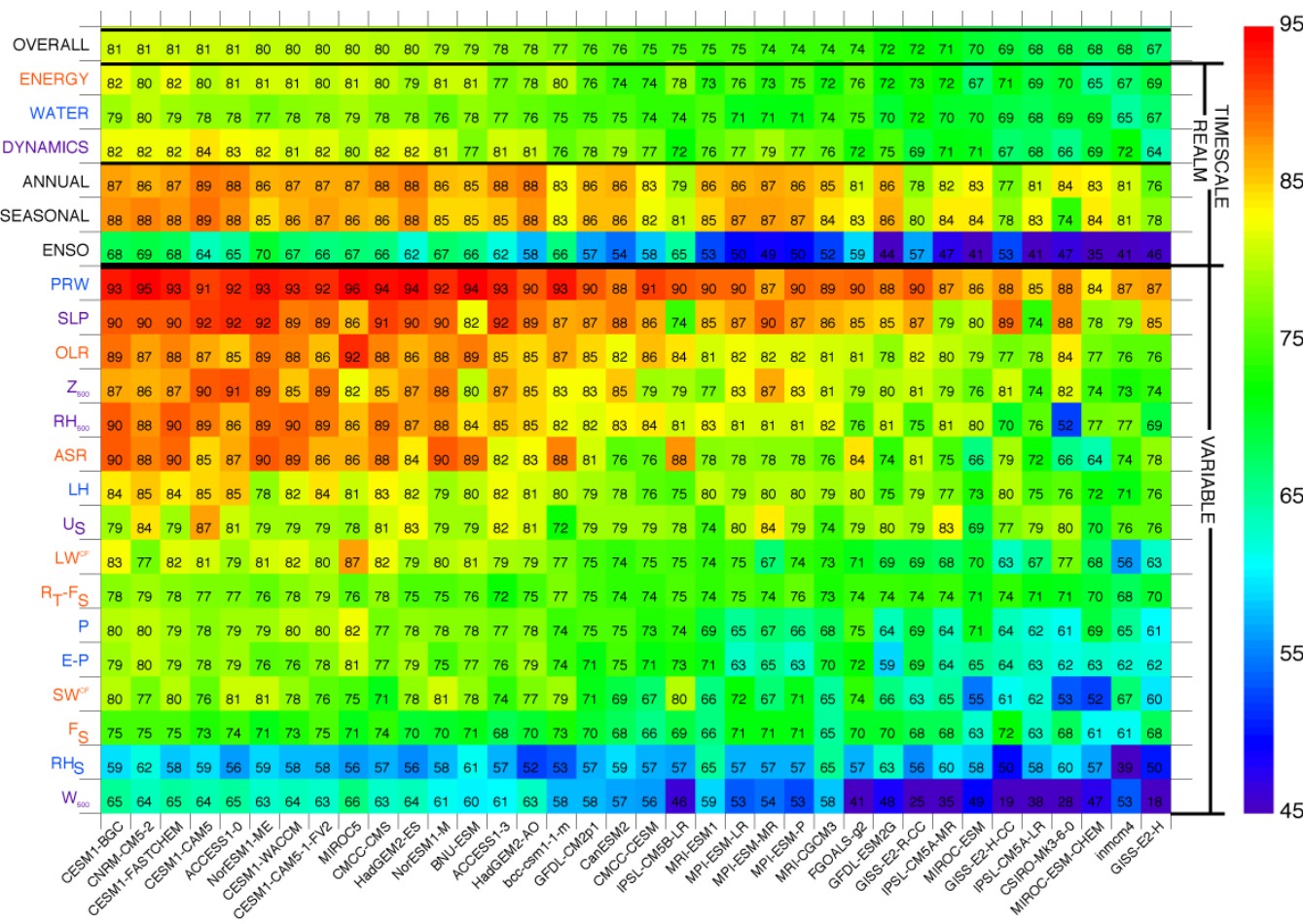

Figure 3: As in Fig. 2 except for historical simulations submitted to the CMIP5 archive.

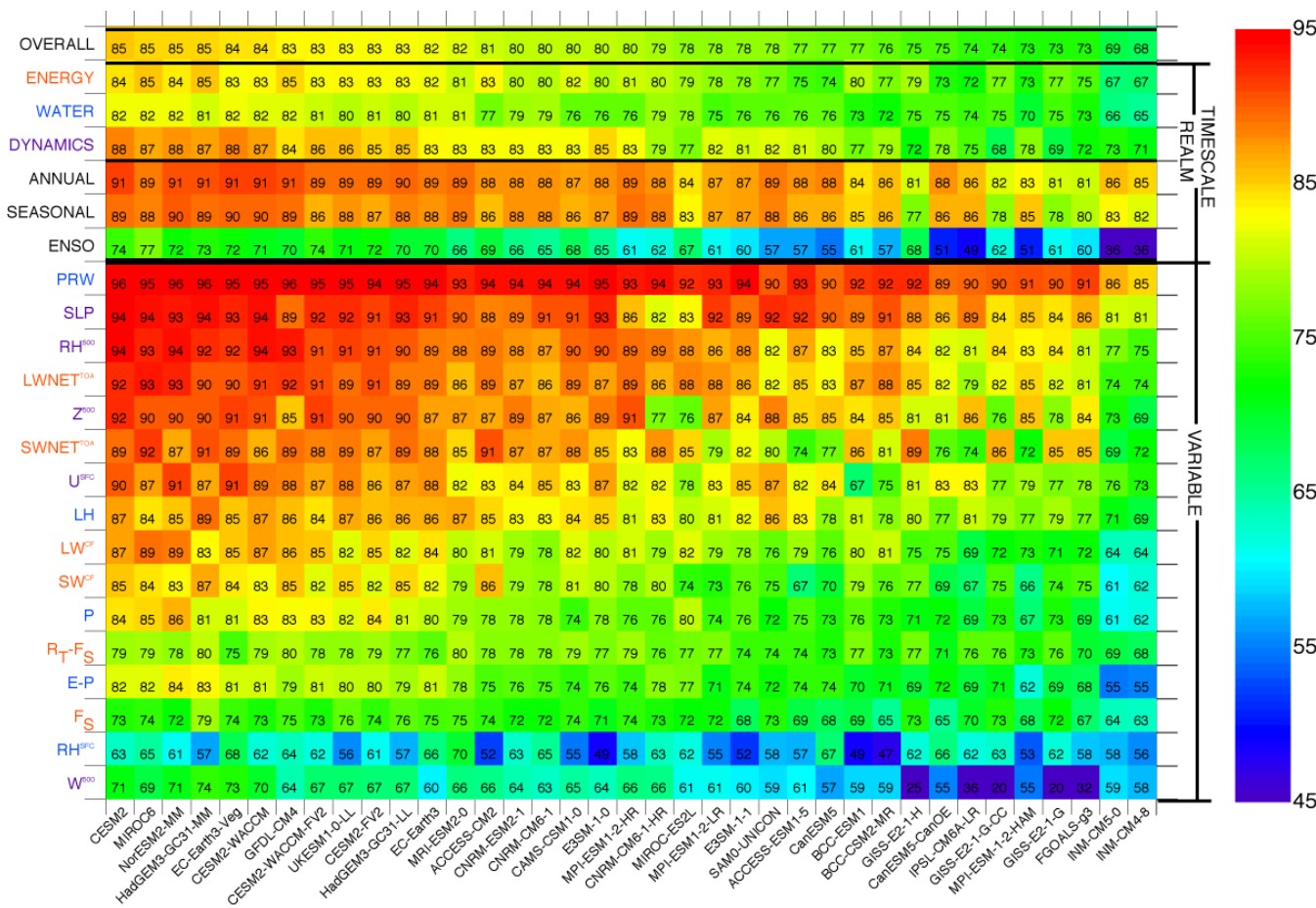

**Figure 4: As in Fig. 2 except for historical simulations submitted to the CMIP6 archive.**

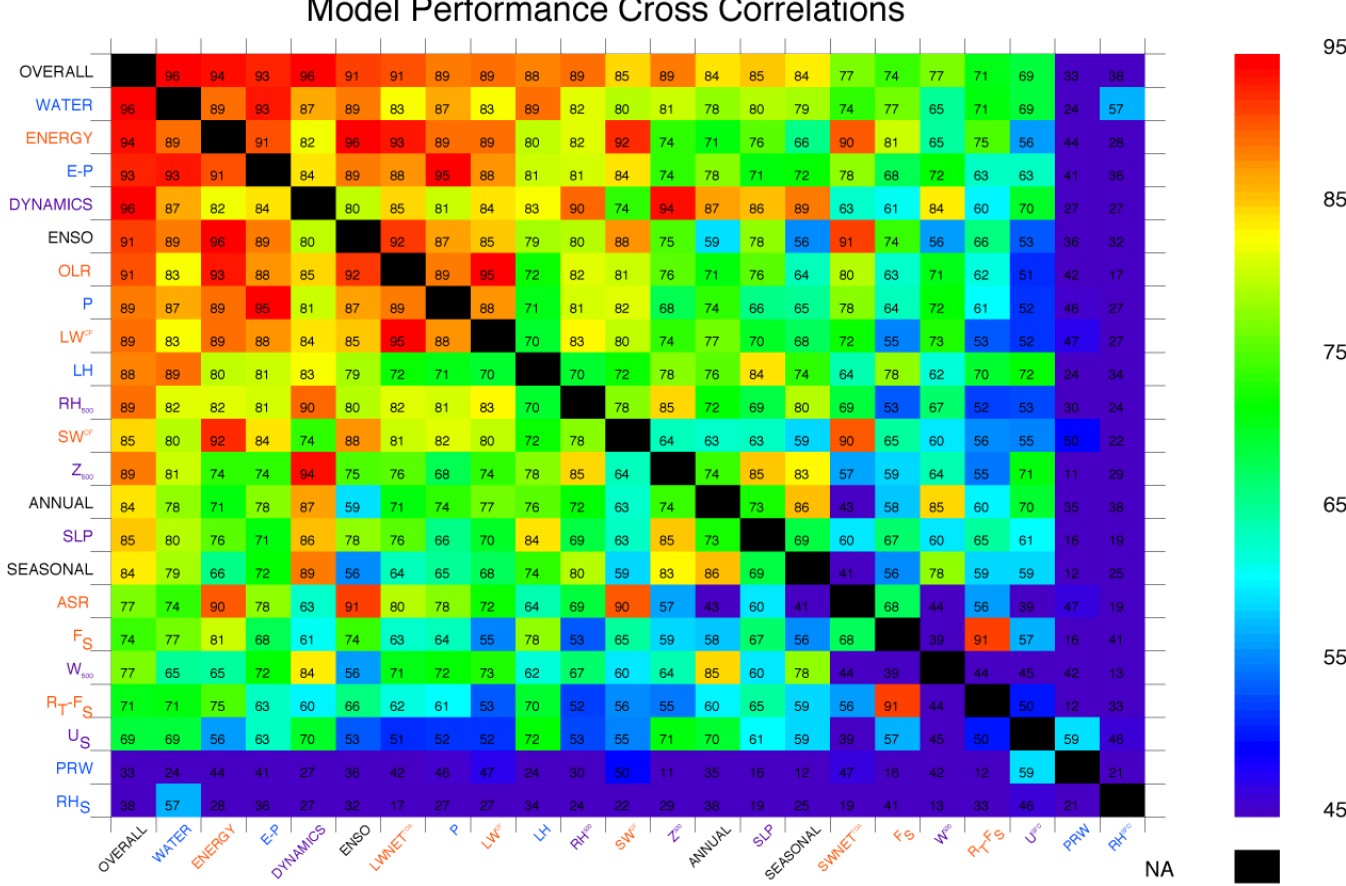

**Figure 5: Cross correlations between variable and aggregate scores computed for the all CMIP archives sorted in order of decreasing correlations from left to right and top to bottom.**

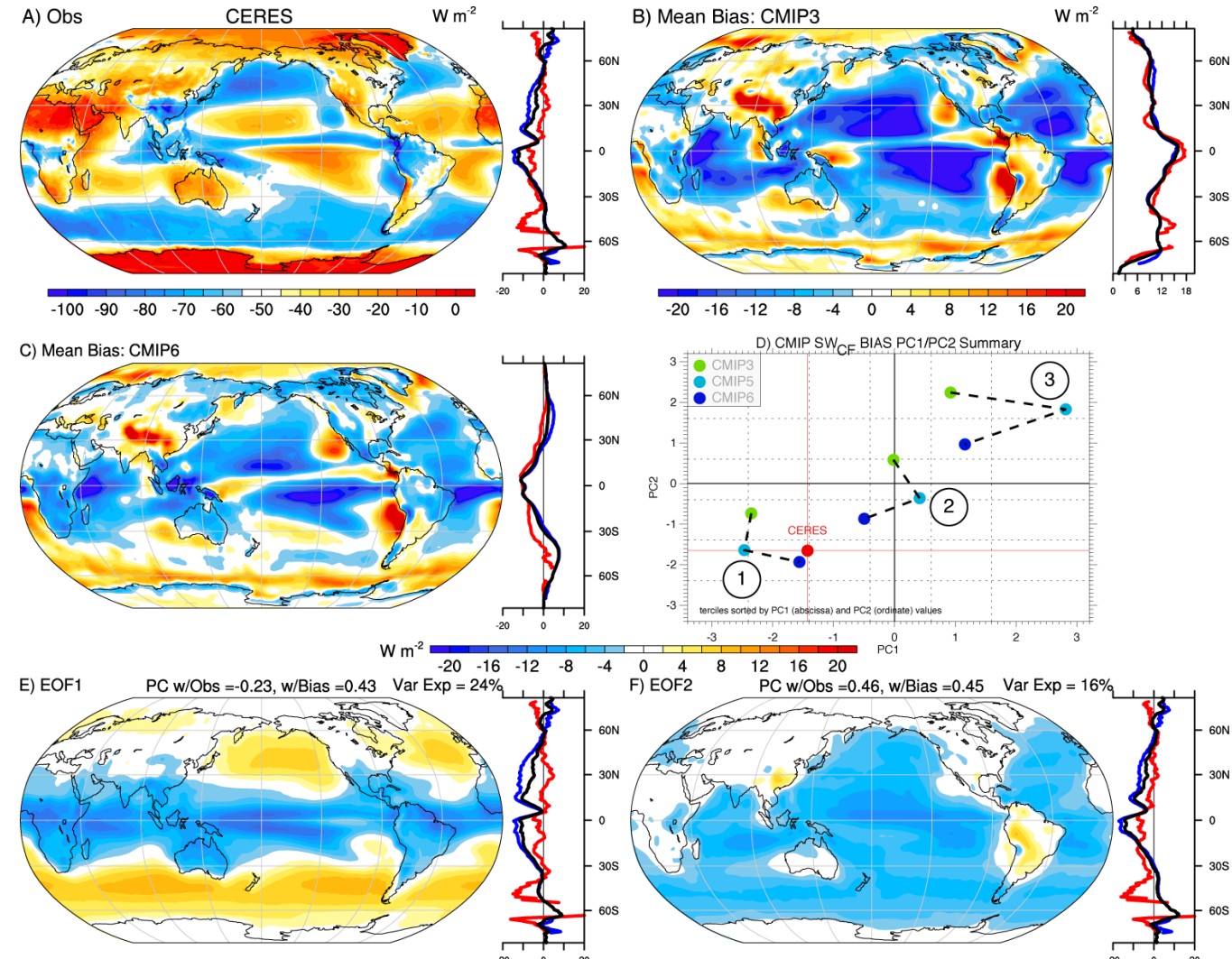

**Figure 6: Analysis of the annual mean SW$_{CF}$ bias in the combined historical CMIP3/5/6 archive including A) the observed estimate from CERES EBAFv4.1, the mean biases in (B) CMIP3 and (C) CMIP6, and (D) the first two PCs of biases and their tercile averages across the CMIP archives, and the associated first (E) and second (F) EOFs of biases. All units are W m$^{-2}$, except for the PCs, which are unitless. Zonal means (right panels) include land (red), ocean (blue), and global (black).**

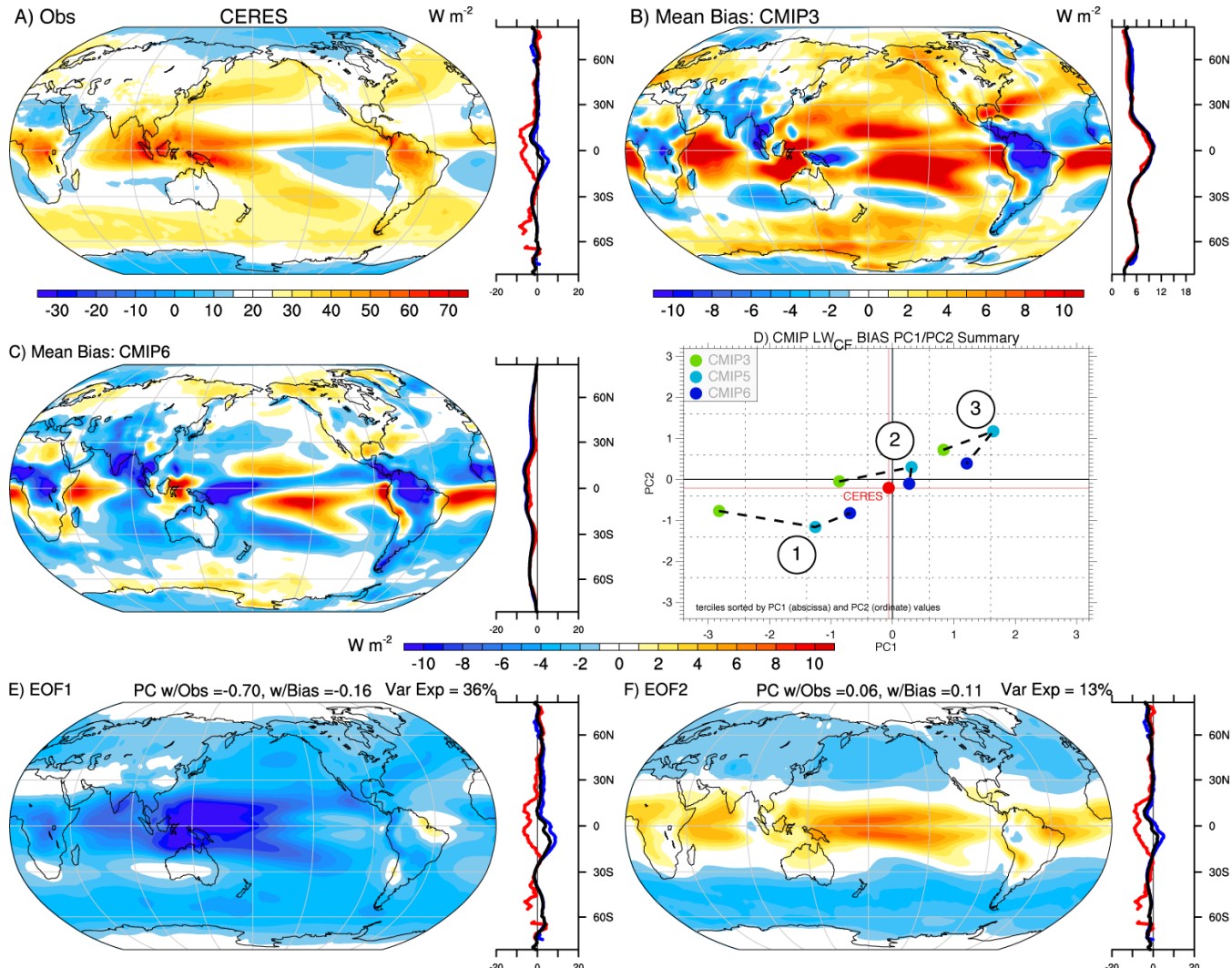

**Figure 7:** Analysis of the annual mean $LW_{CF}$ bias in the combined historical CMIP3/5/6 archive including A) the observed estimate from CERES EBAFv4.1, the mean biases in (B) CMIP3 and (C) CMIP6, and (D) the first two PCs of biases and their tercile averages across the CMIP archives, and the associated first (E) and second (F) EOFs of biases. All units are W m⁻², except for the PCs, which are unitless. Zonal means (right panels) include land (red), ocean (blue), and global (black).

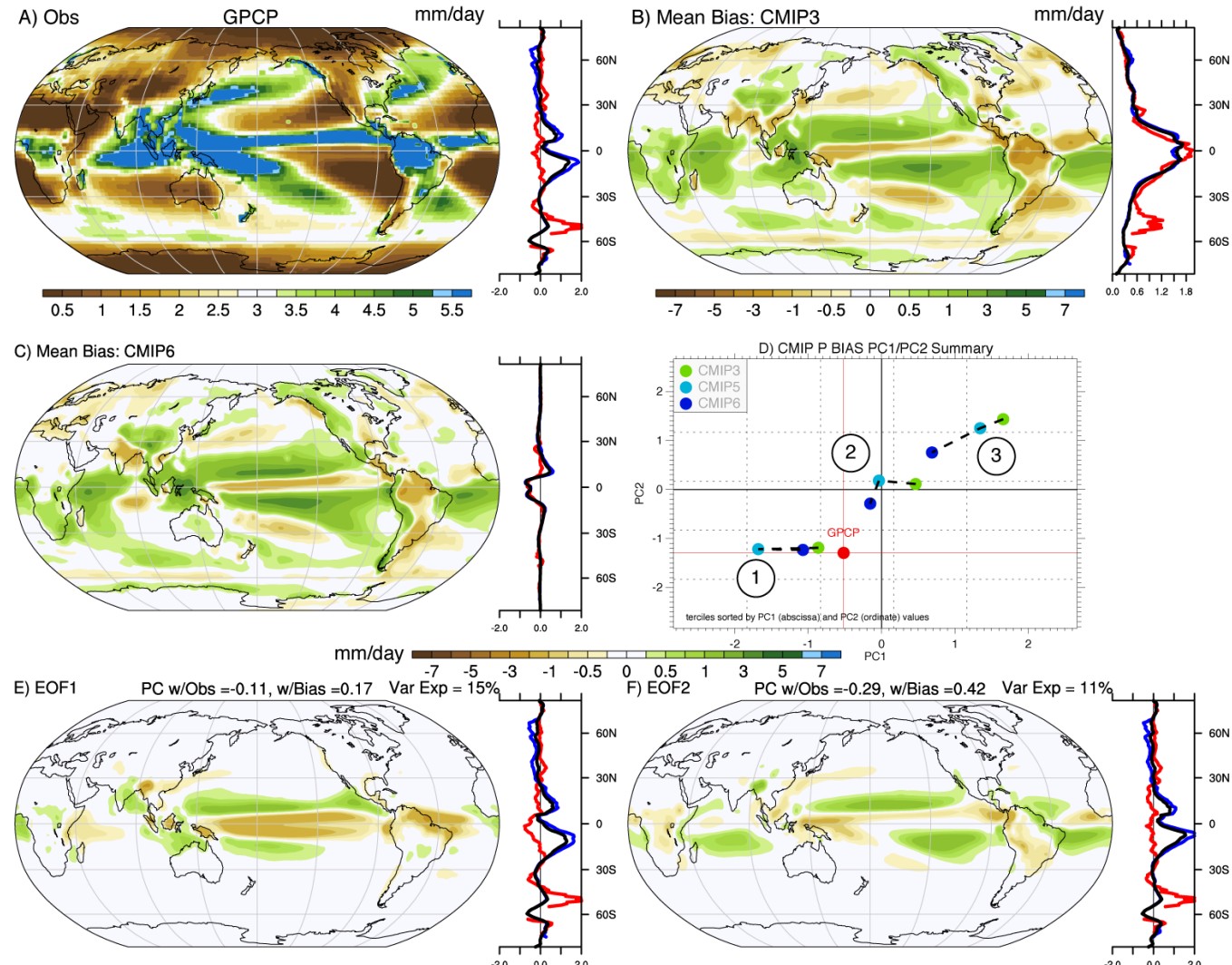

**Figure 8: Analysis of the annual mean precipitation bias in the combined historical CMIP3/5/6 archive including A) the observed estimate from GPCP CDR, the mean biases in (B) CMIP3 and (C) CMIP6, and (D) the first two PCs of biases and their tercile averages across the CMIP archives, and the associated first (E) and second (F) EOFs of biases. All units are mm day⁻¹, except for the PCs, which are unitless. Zonal means (right panels) include land (red), ocean (blue), and global (black).**

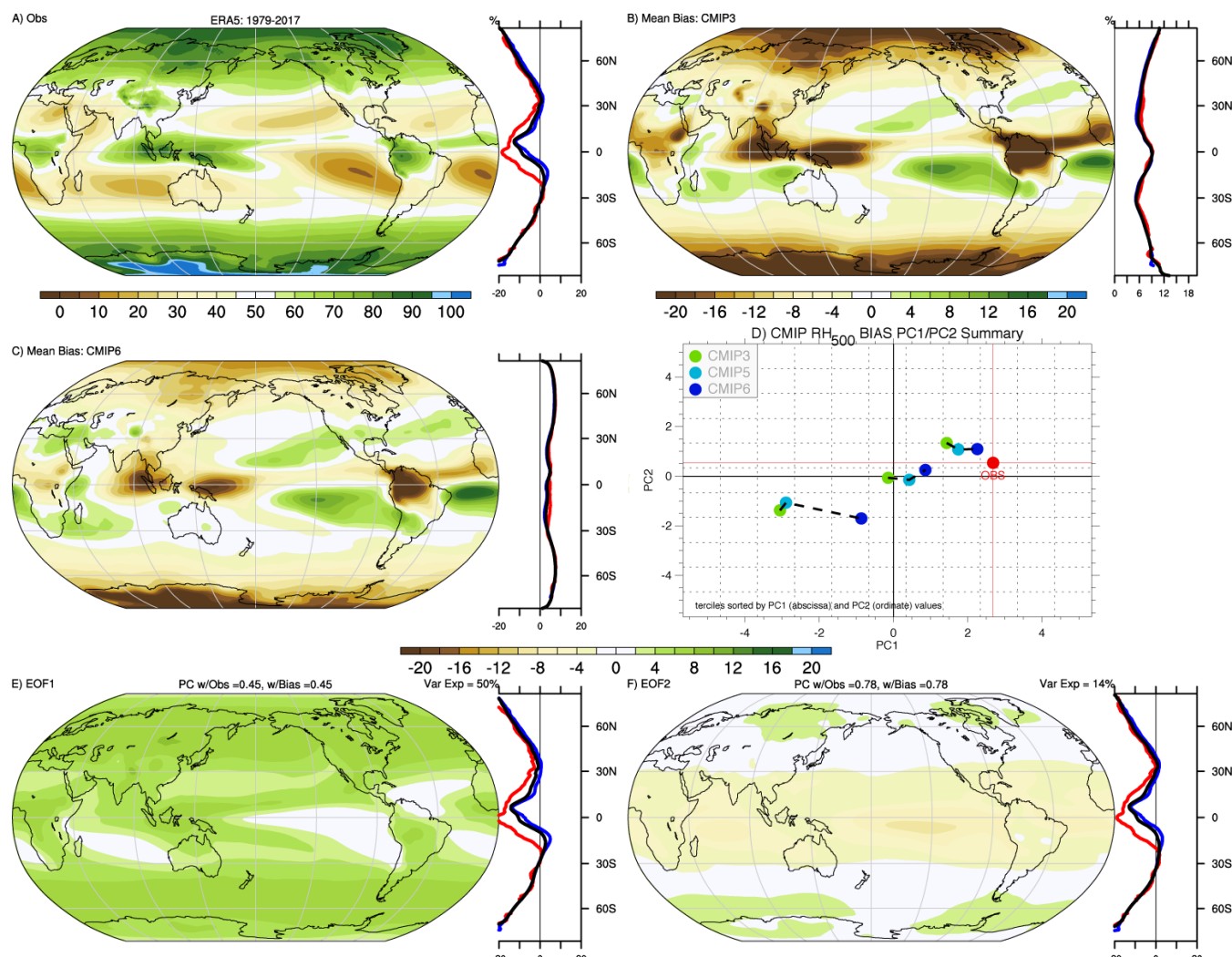

**Figure 9: Analysis of the annual mean RH$_{500}$ bias in the combined historical CMIP3/5/6 archive including A) the observed estimate from ERA5, the mean bias in (B) CMIP3 and (C) CMIP6, and (D) the first two PCs of biases and their tercile averages across the CMIP archives, and the associated first (E) and second (F) EOFs of biases. All units are %, except for the PCs, which are unitless. Zonal means (right panels) include land (red), ocean (blue), and global (black).**

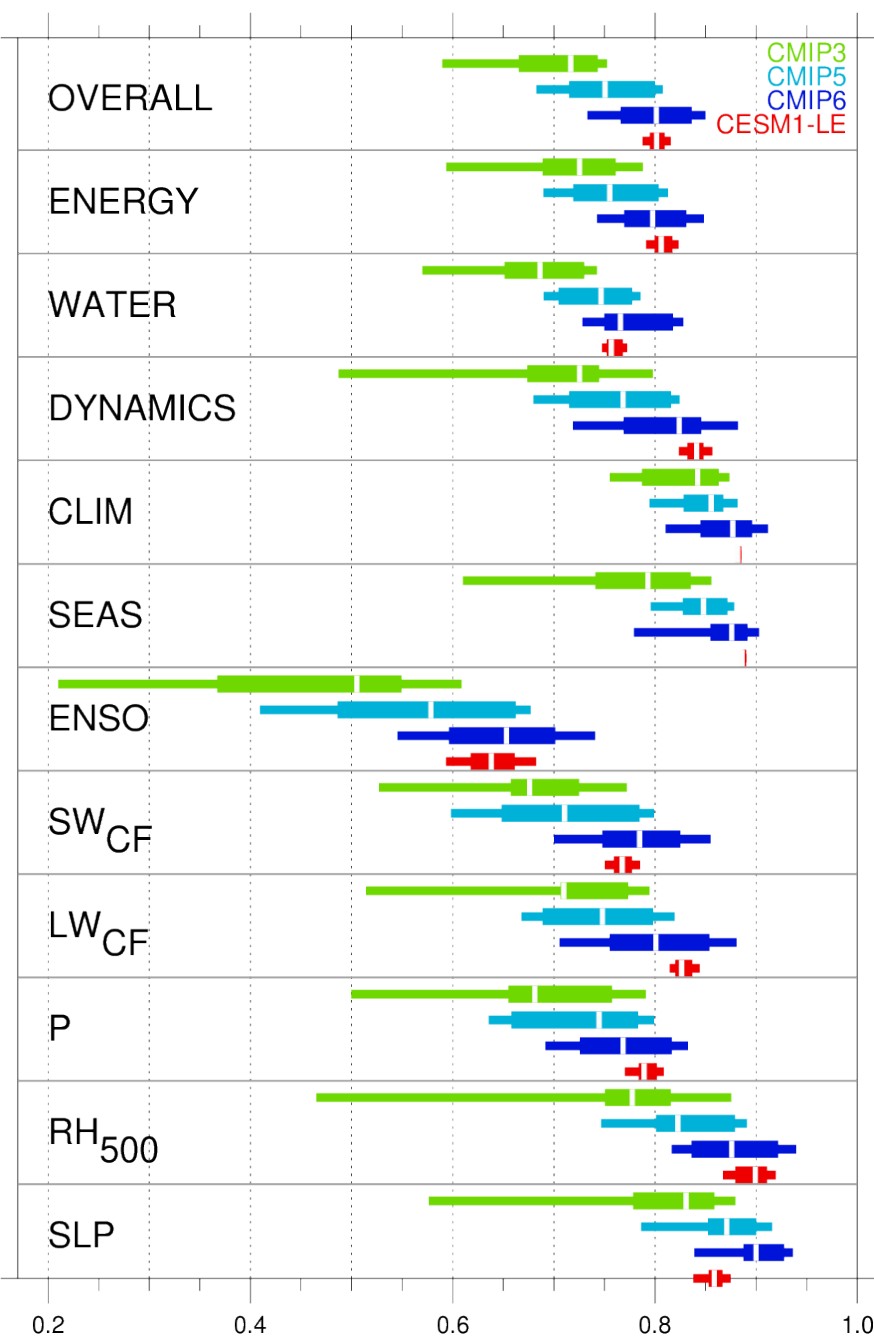

**Figure 10**: Evolution of the distribution of aggregate and selected variable scores across the CMIP archives and the CESM1-LE.