# Peer review of "Evaluating Simulated Climate Patterns from the CMIP Archives Using Satellite and Reanalysis Datasets"

_Geoscientific Model Development, 2020_

## Referee Comment (RC1) · Anonymous Referee #1 · 1 Apr 2020

The manuscript describes an objective approach to evaluate biases in climate model simulations, providing scores based on pattern correlation between key model fields and the most up-to-date observational datasets. Variables are selected on the basis of the most relevant open issues raised on model performances, and are gathered in three realms: the energy budget realm, the water cycle realm, the dynamical realm. Overall scores are obtained, combining weighted scores from different variables, and different timescales are taken into account. The improvement (or lack of) across different generation of the CMIP experiments is also assessed.

Overall, I think that the paper contains some interesting and useful comparisons, and,

as far as I am aware of, it is the first time that such diverse metrics are gathered, in order to assess biases in coupled model simulations in a synthetic and comprehensive way. Extending the analysis to newly available CMIP6 datasets is also a valuable point.

What I found lacking is a bit of context about other model diagnostics and a discussion of the physical relevance of biases. I also have a few remarks about the completeness in describing the methodology. I provide some suggestions on how the paper could be improved in the specific comments below. My general opinion is that the manuscript could be published, subject to minor revisions, as I detail in the following.

————-

Specific comments

l. 52-58: I found that this paragraph, focusing on model diagnostics as a research community service, lacks a bit of context in terms of background on how diagnostics of model performances have been developed in the context of the IPCC and the PCMDI. I also think that this section might benefit of a survey of known sources of biases in models, e.g. the parametrisations, the unresolved scales, the choice of the grids, the numerical scheme. In this respect, the author might mention some of the diagnostics and metrics that have been most recently designed to address some of the specific issues that are considered here, as for example Greve et al. 2018, for the water cycle, precipitation and its regional downscaling, or Lembo et al. 2019, for radiative budgets and transports.

l. 65-67: When data records are not available, I think that it is also important to weigh models beforehand, when the multi-model inter-comparison is performed (e.g. Knutti et al. 2017). These approach has been successfully applied to regional downscaling of global climate model projections (e.g Lorenz et al. 2018), proving that metrics are more relevant to the end user of the model exercise, if models are appropriately weighted. I wonder if it would be possible to adopt a similar approach, with relatively small effort, to the analysis here presented.

l. 68: I think that the appropriate reference for this is Hourdin et al. 2017. Schmidt et al. 2017 refer to a subset of US models from those analysed in Hourdin et al. 2017.

ll. 112-113: I agree that from an observational-based point of view the net surface fluxes are the most challenging, especially if dealing with satellite measurements and inverse techniques. On the other hand, from a model perspective, surface fluxes are the result of several parametrisations and are thus straightforwardly provided, while the retrieval of the vertical integral of atmospheric energy divergence is made difficult by the vertical discretisation and numerical sources of mass imbalance, requiring offline corrections. I think this would be worth mentioning here.

l. 135: why is the 500hPa eddy geopotential height preferred to the 500hPa geopotential height, which is usually made available as an output of a climate model (e.g. in the ESGF repositories for CMIP datasets)? Sect. 2.2: this is the only sentence in the manuscript where the methodology is mentioned. Even though the usage of pattern correlation is a quite usual practice for performance scores, it would be good to have a more detailed description of the method, at least of how the averages are weighted. In general, for sake of clarity, I would suggest to rearrange this first part of the manuscript in order to include a Data and Methods section.

Another suggestion is that the author mentions other possible ways to attribute a performance score to models based on its consistency with observational-based measurements. One can refer, for instance, among others, to the Wasserstein distance, as in Breverman et al. 2017, but there are many other examples...

l. 152: I wonder if there is a non-empirical explanation for the choice of weighting the ENSO timescale less than the annual and seasonal timescales in CESM1-LE.

ll. 155-156: this seems to me a pretty strong assumption, because I see no particular reason why the impact on the overall score from internal variability in other models shall be comparable to the one found in CESM1-LE.

ll. 166-168: a way to test the assumption mentioned in my previous comment could be to focus on a few CMIP models providing a reasonably large ensemble against the CESM1-LE. Would that be feasible?

Sect. 4.0: at this point, the author starts to describe the main results of the analysis. I am a bit puzzled, though, by the fact that no convincing discussion has been provided on the choice of the variables. While for the energy budget and water cycle realm it is clear to me that the author follows from the expert consensus outlined in Burrows et al. 2018, the variables for the dynamical regime seem to me not supported by sufficient argumentation. For instance, why is the eddy geopotential height preferred to the potential vorticity in the free troposphere? If the idea is to meet the experts' needs for key metrics, why not additionally considering the zonal mean wind or the potential vorticity at specific isobaric levels? These variables are fundamental for studies of the atmospheric dynamics, even though they have not been addressed in the paper by Burrows et al. 2018 or, if they are considered, they do not reach a (very) high consensus about their relevance.

ll. 266-268: stated like this, it seems to me more suggesting that only the central tercile is actually closing up to observations across the CMIP generations. . .

ll. 317-319: I wonder if the author might want to comment on why this is the case, and whether this could be really considered as an improvement in the overall performance of the multi-model mean.

l. 325: are these metrics telling something relevant about the behavior of subset of CMIP6 models with high sensitivity. Can something be said about it?

Figure 1: please add in the captions what the blue, red and black meridional sections displayed next to each map describe.

———————————

Technical corrections

l. 36: replace "increasing" by "increasingly".

l. 217: replace "import" with "important".

l. 223: Replace "Select" with "Selected".

ll. 239-241 (and elsewhere): I think that it is sufficient to describe the layout of similar figures only once, when introducing Figure 6 and its panels. Considering removing the introductory sentence in this paragraph and in the successive ones.
* * *
References

Braverman, A., Chatterjee, S., Heyman, M., and Cressie, N.: Probabilistic evaluation of competing climate models, Adv. Stat. Clim. Meteorol. Oceanogr., 3, 93–105, 2017

Greve, P., Gudmundsson, L., and Seneviratne, S. I. Regional scaling of annual mean precipitation and water availability with global temperature change, Earth Syst. Dynam., 9, 227–240, 2018

Hourdin F, Mauritsen T, Gettelman A, et al. The Art and Science of Climate Model Tuning. Bull Am Meteorol Soc 98:589–602, 2017

Knutti, R., Sedláček, J., Sanderson, B. M., Lorenz, R., Fischer, E. M., and Eyring, V. A climate model projection weighting scheme accounting for performance and interdependence, Geophys. Res. Lett., 44, 1909–1918, 2017

Lembo, V., Lunkeit, F., and Lucarini, V.: TheDiaTo (v1.0) – a new diagnostic tool for water, energy and entropy budgets in climate models, Geosci. Model Dev., 12, 3805–3834, 2019

Lorenz, R., Herger, N., Sedláček, J., Eyring, V., Fischer, E. M., & Knutti, R. Prospects and caveats of weighting climate models for summer maximum temperature projections over North America. Journal of Geophysical Research: Atmospheres, 123, 4509–

4526, 2018

---

## Author Comment (AC1) · 22 Apr 2020

Thank you for the time spent reviewing the manuscript and for your constructive comments, which I agree point toward worthwhile improvements in the manuscript.

I have gone through the suggested literature and agree that these and various additional references (cited therein such as Gleckler et al. 2008, Pincus et al. 2008) provide important context. These are now also integrated into the discussion in the manuscript revision. I have also expanded discussion of the methods and expanded on the sources of bias in models. I plan to submit the revision once feedback from Reviewer #2 is posted.

---

## Referee Comment (RC2) · Anonymous Referee #2 · 21 May 2020

The manuscript "Evaluating Simulated Climate Patterns from the CMIP Archives Using Satellite and Reanalysis Datasets" by J.T. Fasullo describes a methodology how developments and improvements of Earth system models can tracked and objectively evaluated using observational datasets and their uncertainties. With the increasing complexity of the models participating in CMIP, a new way of evaluating their proximity to observed parameters is important. While there are already some evaluating and grading methods available, this new method uses some different fields than usual (seasonal differences, ENSO), and also takes into account observational uncertainties. The manuscript is mostly very well written and well structured. However, there are a few things that I think would help to improve the manuscript, and that I would suggest

the author to consider while revising the manuscript. These comments are outlined below.

I therefore recommend the publication of the manuscript after minor revisions.

**General comments:**

- I think it would be helpful to put the method in perspective with other evaluation and grading methodologies (e.g. Gleckler et al., 2008; Reichler and Kim, 2008) that are available for the reader to understand the similarities and differences of the described method to already existing methods.

- I agree with reviewer 1 that the methodology needs to be described in a lot more detail. At the moment it is not clear how the scores are calculated exactly.

- I think it would be helpful to show not just examples for the annual mean bias patterns, but also one of the seasonal patterns and one of the ENSO patterns. After all, these are different to other methods and are therefore definitely worth some more detailed description.

**More specific comments:**

- l. 36: "increasing" -> should be "increasingly"?

- l. 84: Why was the ENSO pattern chosen as one of the bias fields to be evaluated? Could you provide a little more background information about this decision here?

- l. 129: "ERAI" -> should be "ERA-Interim"?

- l. 142-153: This is the that, in my opinion, needs a lot more detail to be easily understandable. How are the scores for the different realms combined from the

individual variable scores? How exactly is the weighting determined? There is a brief example in line 152-153, but even this does not make it clear how the weighting factor was determined.

- l. 155: What does the "0.04" mean? What kind of value range can be expected?

- l. 178-180: Explain the stippling and hatching in a little more detail.

- l. 213: What exactly is cross correlated? All CMIP results at the same time?

- l. 217: "are" -> should be "as"?

- l. 217: "import" -> should be "important"?

- l. 235: I think it would be good to very briefly mention what it means in the plot when the bias diminishes.

- l. 410: Figure 1. What do the three colored lines at the right edge of each global map show? They are not mentioned or explained anywhere.

- l. 413: "CESM-CERES differences exceed twice the estimated internal spread" -> this seems slightly different to the definition presented in the first paragraph of Section 3. Please adjust this so that it is clearer and the same in both parts.

- l. 434: Figure 6. What do the colored lines to the right of the global maps represent in this figure?

**References:**

- Gleckler et al., JGR, 2008, doi:10.1029/2007jd008972

- Reichler and Kim, BAMS, 2008, doi:10.1175/bams-89-3-303

---

## Author Comment (AC2) · 26 May 2020

Thank you for the time spent reviewing the manuscript and your constructive suggestions for improvement. Many of your remarks echo the suggestions of Rev 1 and I find merit in both, particularly in regards to contextualizing my approach relative to previous efforts and adding to the discussion of methodological details. I also find value in expanding the discussion of why ENSO patterns are considered as it helps underscore an important point regarding the motivations for the analysis - the assessment of the coupled behavior of models. Also thank you for feedback on some of the minor details in the manuscript. I believe I will be able to fully address the comments made in a

revision.
* * *

---

## Author Response (AR1)

The manuscript describes an objective approach to evaluate biases in climate model simulations, providing scores based on pattern correlation between key model fields and the most up-to-date observational datasets. Variables are selected on the basis of the most relevant open issues raised on model performances, and are gathered in three realms: the energy budget realm, the water cycle realm, the dynamical realm. Overall scores are obtained, combining weighted scores from different variables, and different timescales are taken into account. The improvement (or lack of) across differ- ent generation of the CMIP experiments is also assessed.

Overall, I think that the paper contains some interesting and useful comparisons, and, as far as I am aware of, it is the first time that such diverse metrics are gathered, in order to assess biases in coupled model simulations in a synthetic and comprehensive way. Extending the analysis to newly available CMIP6 datasets is also a valuable point.

What I found lacking is a bit of context about other model diagnostics and a discussion of the physical relevance of biases. I also have a few remarks about the completeness in describing the methodology. I provide some suggestions on how the paper could be improved in the specific comments below. My general opinion is that the manuscript could be published, subject to minor revisions, as I detail in the following.

**Thank you for the time spent reviewing the manuscript and for your constructive comments, which I agree point toward worthwhile improvements in the manuscript.**

**I have gone through the suggested literature and agree that these and various additional references (cited therein such as Gleckler et al. 2008, Pincus et al. 2008) provide important context. Discussion of these references is now integrated into the manuscript revision. I also emphasize that the advance of the current work lies in 1) its consideration of 3 CMIP generations, 2) its quantification of Internal variability on computed scores, 3) its use of more advanced and insightful fields, and 4) its consideration of recent expert elicitations on the fields that are key to model evaluation. In response to the concerns above, I have also considerably expanded discussion of the methods and expanded on the sources of bias in models, though that remains an area of active research.**

————-

**Specific comments**

65  l. 52-58: I found that this paragraph, focusing on model diagnostics as a research community service, lacks a bit of context in terms of background on how diagnostics of model performances have been developed in the context of the IPCC and the PCMDI. I also think that this section might benefit of a survey of known sources of biases in models, e.g. the parametrisations, the unresolved scales, the choice of the grids, the numerical scheme. In this

70  respect, the author might mention some of the diagnostics and metrics that have been most recently designed to address some of the specific issues that are considered here, as for example Greve et al. 2018, for the water cycle, precipitation and its regional downscaling, or Lembo et al. 2019, for radiative budgets and transports.

75  **lines 52-58: I agree that the manuscript would benefit for an enhancement of this kind of context. Toward this end I now include discussion of substantially more literature.**

l. 65-67: When data records are not available, I think that it is also important to weigh models beforehand, when the multi-model inter-comparison is performed (e.g. Knutti et al. 2017).

80  These approach has been successfully applied to regional downscaling of global climate model projections (e.g Lorenz et al. 2018), proving that metrics are more relevant to the end user of the model exercise, if models are appropriately weighted. I wonder if it would be possible to adopt a similar approach, with relatively small effort, to the analysis here presented.

85  **lines 65-67: I am a bit unclear as to what is being suggested here since our focus is global rather than regional. The Lorenz paper develops a weighting scheme for a specific regional application (temperature projections over North America). Given that there is no analogous targeted application here, it doesn't seem that such a weighting**

90  **scheme would be appropriate. That said, the potential use of the model scores generated in this work may be of use to targeting applications and this is part of the reason for the distribution of datasets. Associated discussion has been added to highlight this.**

95  l. 68: I think that the appropriate reference for this is Hourdin et al. 2017. Schmidt et al. 2017 refer to a subset of US models from those analysed in Hourdin et al. 2017.

**line 68: Thank you, I agree. The Hourdin references is now added.**

100  ll. 112-113: I agree that from an observational-based point of view the net surface fluxes are the most challenging, especially if dealing with satellite measurements and inverse techniques. On the other hand, from a model perspective, surface fluxes are the result of several parametrisations and are thus straightforwardly provided, while the retrieval of the

vertical integral of atmospheric energy divergence is made difficult by the vertical discretisation and numerical sources of mass imbalance, requiring offline corrections. I think this would be worth mentioning here.

**lines 112-113: Yes, the additional uncertainty of observed radiative fluxes means that the signal of model bias in surface fluxes is not nearly as large relative to uncertainty as it is for TOA. So long as atmospheric model components conserve energy well (relative to their biases, a condition that we find to be met for CMIP models), the vertical integral can reasonably be inferred from the TOA minus surface budgets (though lack of closure may exist in other components). This skirts the issue of offline calculations raised by the reviewer.**

l. 135: why is the 500hPa eddy geopotential height preferred to the 500hPa geopotential height, which is usually made available as an output of a climate model (e.g. in the ESGF repositories for CMIP datasets)? Sect. 2.2: this is the only sentence in the manuscript where the methodology is mentioned. Even though the usage of pattern correlation is a quite usual practice for performance scores, it would be good to have a more detailed description of the method, at least of how the averages are weighted. In general, for sake of clarity, I would suggest to rearrange this first part of the manuscript in order to include a Data and Methods section.

**line 135: The pattern correlation of eddy geopotential height (rather than geopotential height) is more of a challenge for modes and a better indicator of the dynamic flows that we are trying to diagnose, since the zonal mean temperature component is rather mundane yet can overwhelm the spatial variance of geopotential. Additional motivation for the selection of variables, including 500 hPa eddy geopotential (readily derived from removing the zonal mean of 500 hPa geopotential) has been added.**

Another suggestion is that the author mentions other possible ways to attribute a performance score to models based on its consistency with observational-based mea- surements. One can refer, for instance, among others, to the Wasserstein distance, as in Breverman et al. 2017, but there are many other examples...

**line 152: I don't find explicit mention of the Wasserstein Distance in the Braverman et al. manuscript. Perhaps the reviewer is referring to the distance "DI" defined in line 162 of their manuscript? In the spirit of this suggestion, considerable additional discussion of other scoring approaches is now added.**

l. 152: I wonder if there is a non-empirical explanation for the choice of weighting the ENSO timescale less than the annual and seasonal timescales in CESM1-LE.

**No, this choice is motivated purely by the desire to have a readily interpretable influence of internal variability in the overall scores, which is deemed to be very important.**

ll. 155-156: this seems to me a pretty strong assumption, because I see no particular reason why the impact on the overall score from internal variability in other models shall be comparable to the one found in CESM1-LE.

**It is true that other models may have differing strengths of internal variability. That said, this is the first attempt we know of to score models with consideration of such influence. Future work will seek to improve this, though doing so will depend on multiple model realizations (not all CMIP6 simulations even meet this threshold).**

ll. 166-168: a way to test the assumption mentioned in my previous comment could be to focus on a few CMIP models providing a reasonably large ensemble against the CESM1-LE. Would that be feasible?

**Yes, as alluded to in the manuscript the analysis of several large ensembles has been performed with results posted online. We find the CESM1-LE to do a reasonable job at estimating the range of internal variability. Including multiple large ensembles in the present manuscript does not change much and introduces a layer of confusion arguably that detracts from the work. We do hope to address this further in the future however.**

Sect. 4.0: at this point, the author starts to describe the main results of the analysis. I am a bit puzzled, though, by the fact that no convincing discussion has been provided on the choice of the variables. While for the energy budget and water cycle realm it is clear to me that the author follows from the expert consensus outlined in Burrows et al. 2018, the variables for the dynamical regime seem to me not supported by sufficient argumentation. For instance, why is the eddy geopotential height preferred to the potential vorticity in the free troposphere? If the idea is to meet the experts' needs for key metrics, why not additionally considering the zonal mean wind or the potential vorticity at specific isobaric levels? These variables are fundamental for studies of the atmospheric dynamics, even though they have not been addressed in the paper by Burrows et al. 2018 or, if they are considered, they do not reach a (very) high consensus about their relevance.

**We acknowledge that the choice of variables has a subjective component. Our choice has been motivated in part by the feedback of modelers at various modeling centers**

**including NCAR. Note that Burrow et al. 2018 does cover SLP, which is one of our dynamic fields.**

ll. 266-268: stated like this, it seems to me more suggesting that only the central tercile is actually closing up to observations across the CMIP generations. . .

**In PC1 and PC2 space, CMIP6 terciles lie closer to GPCP in terciles 2/3 than CMIP3/5. In tercile 1, CMIP3 is slightly closer than CMIP6. Related discussion has been added to the text.**

ll. 317-319: I wonder if the author might want to comment on why this is the case, and whether this could be really considered as an improvement in the overall performance of the multi-model mean.

**Added: " In part this may be due to the elimination of very low resolution models in CMIP5/6, though improvements in model physics is also likely to play a role. "**

l. 325: are these metrics telling something relevant about the behavior of subset of CMIP6 models with high sensitivity. Can something be said about it?

**Given the limited degrees of freedom, we would rather await the completion of the CMIP6 simulations before speculating on this.**

Figure 1: please add in the captions what the blue, red and black meridional sections displayed next to each map describe.

**Added. Thank you.**

——————— **Technical corrections**
 l. 36: replace "increasing" by "increasingly". l. 217: replace "import" with "important".

**Replaced. Thank you.**

l. 223: Replace "Select" with "Selected".

**Replaced. Thank you.**

ll. 239-241 (and elsewhere): I think that it is sufficient to describe the layout of similar figures only once, when introducing Figure 6 and its panels. Considering removing the introductory sentence in this paragraph and in the successive ones.

**I have condensed successive figure introductions rather than eliminate them entirely as I feel some context is needed.**

—————– **References**

Braverman, A., Chatterjee, S., Heyman, M., and Cressie, N.: Probabilistic evaluation of competing climate models, Adv. Stat. Clim. Meteorol. Oceanogr., 3, 93–105, 2017

Greve, P., Gudmundsson, L., and Seneviratne, S. I. Regional scaling of annual mean precipitation and water availability with global temperature change, Earth Syst. Dy- nam., 9, 227–240, 2018

Hourdin F, Mauritsen T, Gettelman A, et al. The Art and Science of Climate Model Tuning. Bull Am Meteorol Soc 98:589–602, 2017

Knutti, R., Sedláček, J., Sanderson, B. M., Lorenz, R., Fischer, E. M., and Eyring, V. A climate model projection weighting scheme accounting for performance and interde- pendence, Geophys. Res. Lett., 44, 1909–1918, 2017

Lembo, V., Lunkeit, F., and Lucarini, V.: TheDiaTo (v1.0) – a new diagnostic tool for water, energy and entropy budgets in climate models, Geosci. Model Dev., 12, 3805– 3834, 2019

Lorenz, R., Herger, N., Sedláček, J., Eyring, V., Fischer, E. M., & Knutti, R. Prospects and caveats of weighting climate models for summer maximum temperature projections over North America. Journal of Geophysical Research: Atmospheres, 123, 4509–4526, 2018

**\*\*\*Reply to RC2\*\*\***
The manuscript "Evaluating Simulated Climate Patterns from the CMIP Archives Using Satellite and Reanalysis Datasets" by J.T. Fasullo describes a methodology how developments and improvements of Earth system models can tracked and objectively evaluated using observational datasets and their uncertainties. With the increasing complexity of the models participating in CMIP, a new way of evaluating their proximity to observed parameters is important. While there are already some evaluating and grading methods available, this new method uses some different fields than usual (seasonal differences, ENSO), and also takes into account observational uncertainties. The manuscript is mostly very well written and well structured. However, there are a few things that I think would help to improve the manuscript, and that I would suggest the author to consider while revising the manuscript. These comments are outlined below.

**Thank you for the time spent reviewing the manuscript and your constructive suggestions for improvement.**

I therefore recommend the publication of the manuscript after minor revisions.

General comments:

• I think it would be helpful to put the method in perspective with other evaluation and grading methodologies (e.g. Gleckler et al., 2008; Reichler and Kim, 2008) that are available for the reader to understand the similarities and differences of the described method to already existing methods.

**Thank you - per this and the comments of Rev 1 significant new discussion of previous work has been added.**

• I agree with reviewer 1 that the methodology needs to be described in a lot more detail. At the moment it is not clear how the scores are calculated exactly.

**Thank you - per this and the comments of Rev 1 significant new description of methods has been added.**

• I think it would be helpful to show not just examples for the annual mean bias patterns, but also one of the seasonal patterns and one of the ENSO patterns. After all, these are different to other methods and are therefore definitely worth some more detailed description.

**Figure 1 shows such patterns in the middle and left columns, respectively.**

More specific comments:

• l. 36: "increasing" -> should be "increasingly"?

**Fixed. Thank you.**

• l. 84: Why was the ENSO pattern chosen as one of the bias fields to be evalu- ated? Could you provide a little more background information about this decision here?

**Discussion added.**

• l. 129: "ERAI" -> should be "ERA-Interim"?

**Fixed. Thank you.**

• l. 142-153: This is the that, in my opinion, needs a lot more detail to be easily understandable. How are the scores for the different realms combined from the individual

variable scores? How exactly is the weighting determined? There is a brief example in line 152-153, but even this does not make it clear how the weighting factor was determined.

**The discussion is expanded and clarified.**

• l. 155: What does the "0.04" mean? What kind of value range can be expected?

**This is just the +- 2 sigma range, admittedly a conservative bound.**

• l. 178-180: Explain the stippling and hatching in a little more detail.

**Added. Thank you.**

• l. 213: What exactly is cross correlated? All CMIP results at the same time?

**Reworded.**

• l. 217: "are" -> should be "as"?

**Fixed. Thank you.**

• l. 217: "import" -> should be "important"?

**Fixed. Thank you.**

• l. 235: I think it would be good to very briefly mention what it means in the plot when the bias diminishes.

**Mentioned.**

• l. 410: Figure 1. What do the three colored lines at the right edge of each global map show? They are not mentioned or explained anywhere.

**Their meaning is now mentioned. Thank you.**

• l. 413: "CESM-CERES differences exceed twice the estimated internal spread" -> this seems slightly different to the definition presented in the first paragraph of Section 3. Please adjust this so that it is clearer and the same in both parts.

**Changed. Thank you.**

• l. 434: Figure 6. What do the colored lines to the right of the global maps represent in this figure?

345

**Zonal means - discussion now added. Thank you.**

[revised manuscript text omitted]

---

## Author Response (AR2)

**National Center for Atmospheric Research**
**Climate and Global Dynamics Division**
**Climate Analysis Section**

*Dr. John T. Fasullo*
*fasullo@ucar.edu, http://www.cgd.ucar.edu/staff/fasullo/index.html*
*P. O. Box 3000 ● Boulder, CO  80301*
*Tel: 303-497-1712 ● Fax: 303-497-1333*

**29 Jun 2020**

5  **To:** *Geophysical Model Development*

**From: Dr. John Fasullo**

**Subject: Manuscript Revision**

Dear Editor,

I would like to thank both you and the referee for the time spent in evaluating my manuscript.
Following consideration of these comments, I have submitted my revised manuscript, "Evaluating
15  Simulated Climate Patterns from the CMIP Archives Using Satellite and Reanalysis Datasets using the
Climate Model Assessment Tool (CMATv1)", which includes the suggested change in title and
citations. A copy of the IDL code has also been posted on Zenodo in accordance with the GMD Code
and Data Policy.

20  Sincerely,

John Fasullo

[revised manuscript text omitted]